# Human MicroRNAs Attenuate the Expression of Immediate Early Proteins and HCMV Replication during Lytic and Latent Infection in Connection with Enhancement of Phosphorylated RelA/p65 (Serine 536) That Binds to MIEP

**DOI:** 10.3390/ijms23052769

**Published:** 2022-03-02

**Authors:** Yeon-Mi Hong, Seo Yeon Min, Dayeong Kim, Subin Kim, Daekwan Seo, Kyoung Hwa Lee, Sang Hoon Han

**Affiliations:** 1Division of Infectious Disease, Department of Internal Medicine, Yonsei University College of Medicine, Seoul 06273, Korea; yeonmih@yuhs.ac (Y.-M.H.); kyungstella@gmail.com (S.Y.M.); dayoung747@yuhs.ac (D.K.); subink93@yuhs.ac (S.K.); khlee0309@yuhs.ac (K.H.L.); 2Severance Biomedical Science Institute, Yonsei University College of Medicine, Seoul 06273, Korea; daegwanseo@yuhs.ac

**Keywords:** HCMV, microRNA, immediate early protein, latent infection, MIEP, NF-κB, RelA/p65, phosphorylation, serine 536

## Abstract

Attenuating the expression of immediate early (IE) proteins is essential for controlling the lytic replication of human cytomegalovirus (HCMV). The human microRNAs (hsa-miRs), miR-200b-3p and miR-200c-3p, have been identified to bind the 3′-untranslated region (3′-UTR) of the mRNA encoding IE proteins. However, whether hsa-miRs can reduce IE72 expression and HCMV viral load or exhibit a crosstalk with the host cellular signaling machinery, most importantly the NF-κB cascade, has not been evaluated. In this study, argonaute-crosslinking and immunoprecipitation-seq revealed that miR-200b-3p and miR-200c-3p bind the 3′-UTR of *UL123*, which is a gene that encodes IE72. The binding of these miRNAs to the 3′-UTR of *UL123* was verified in transfected cells stably expressing GFP. We used miR-200b-3p/miR-200c-3p mimics to counteract the downregulation of these miRNA after acute HCMV infection. This resulted in reduced IE72/IE86 expression and HCMV VL during lytic infection. We determined that IE72/IE86 alone can inhibit the phosphorylation of RelA/p65 at the Ser^536^ residue and that p-Ser^536^ RelA/p65 binds to the major IE promoter/enhancer (MIEP). The upregulation of miR-200b-3p and miR-200c-3p resulted in the phosphorylation of RelA/p65 at Ser^536^ through the downregulation of IE, and the binding of the resultant p-Ser^536^ RelA/p65 to MIEP resulted in a decreased production of pro-inflammatory cytokines. Overall, miR-200b-3p and miR-200c-3p—together with p-Ser^536^ RelA/p65—can prevent lytic HCMV replication during acute and latent infection

## 1. Introduction

Lifelong latent infection with human cytomegalovirus (HCMV) can alter the host immune system through the excessive clonal inflation of highly differentiated HCMV-specific CD45RA^+^CD8^+^ effector memory T lymphocytes [1,2]. Subclinical immune senescence caused by the intermittent reactivation of latent HCMV infection can induce various chronic inflammatory morbidities and aging in individuals, causing difficulties in making a good diagnosis and that metagenomics can help in the diagnostic process [1,2,3,4,5,6,7]. Moreover, lytic HCMV replication can be—directly as well as indirectly—detrimental to the health of hematopoietic stem cell or solid organ transplant (SOT) recipients [8,9]. Such crucial effects of the HCMV latency on various populations indicate that primary infection should be controlled or antiviral strategies should be developed based on a thorough understanding of the pathophysiologic mechanisms [10,11,12,13].

The complex processes involved in virion production from latent-to-lytic switch are sequentially coordinated by the expression of major immediate-early (MIE), early, and late genes [5,13,14,15]. The expression of immediate-early (IE) proteins (IE72 encoded by *UL123* and IE86 encoded by *UL122*) immediately after chromatin-mediated epigenetic remodeling of the HCMV DNA is essential for the transcription of early and late genes [5,13,14,15,16,17,18]. While the robust activation of the MIE gene initiates significantly productive HCMV replication, the absence or low levels of IE proteins are directly connected to the establishment of latency in undifferentiated CD34^+^ hematopoietic progenitor cells (HPCs) [12,13,19,20]. Therefore, the sustained downregulation of IEs might be a key element in maintaining latency and preventing the development of a wide range of diseases [13,15,21,22,23].

Attention has been focused on non-coding small interfering RNAs that target MIE genes and microRNAs that inhibit IE expression to understand the pathophysiology of HCMV infection and virus–host interactions during HCMV latency or reactivation as well as to identify novel anti-HCMV agents [13,24,25,26,27,28,29]. Mature hsa-miRs play a central role in inhibiting protein synthesis by binding to the 3′-untranslated region (3′-UTR) of specific mRNA and subsequently targeting them for degradation [30,31,32]. Therapeutics based on miRNAs have novel and promising applications in many inflammatory or degenerative diseases and cancers [31,33]. Several HCMV-encoded miRNAs have unique features, and some hsa-miRs are related to HCMV latency or replication in various pathophysiological processes [24,34,35,36,37,38,39]. 

O’Connor et al. predicted the binding of miR-200b-3p and miR-200c-3p to the 3′-UTR of *UL122* mRNA and subsequent downregulation of IE86 in cells permissive for HCMV replication [34]. We found a decreased expression of miR-200b-3p and miR-200c-3p in pre-transplant peripheral mononuclear cells stimulated using a laboratory HCMV strain in recipients with post-SOT HCMV infection or disease and in HCMV-infected lytic end-organ tissues [35,40]. One study has found a significantly decreased expression of miR-200b-3p and miR-200c-3p in HCMV AD169-infected fibroblasts compared with that in mock-treated cells [41]. In contrast, other studies found that miR-200b-3p and miR-200c-3p are not differentially regulated in cells latently infected with HCMV [28,42]. These findings support the hypothesis that these miRNAs can contribute to the regulation of acute lytic replication and latency, but details of crosstalk virus–host interactions and cellular signaling pathways remain unknown. 

Nuclear factor light-chain-enhancer of activated *B* cells (NF-κB) is rapidly activated and binds to the MIE promoter (MIEP)—a potent transactivator element of genes encoding the IE proteins that harbors four NF-κB-binding motifs—during early productive HCMV replication. NF-κB also regulated the expression of other viral genes such as *UL144* [13,14,43,44,45,46,47]. Rapid induction of the NF-κB signaling pathway in response to HCMV entry is essential for the induction of IE expression, the lytic cascade of progeny virus production, and reactivation of latently infected undifferentiated cells [45,46,48,49,50,51]. This initial increase in nuclear NF-κB activity can be attributed to the release of cytoplasmic interacting partners of NF-κB, i.e., RelA/p65 and p50, and results in the sustained expression of IE genes [44,45]. Phosphorylation of the RelA/p65 domain at the serine 536 (Ser^536^) residue significantly impacts the stability and protein interactions of NF-κB via a conformational change, and it enhances NF-κB transactivation in response to infection with microbes [52,53,54,55,56,57], but this is not known in HCMV infection. Moreover, changes in the NF-κB signaling pathway induced by cellular hsa-miRs during HCMV infection remain unknown. Therefore, we explored the effects of changes in hsa-miRs during HCMV infection on the phosphorylation of RelA/p65 (Ser^536^) to understand how cellular miRNAs are involved in modulating host signaling after HCMV entry.

A single transcript generated from five exons located in the MIE locus comprising *UL122* and *UL123* encodes IE72 and IE86 mRNAs that share 85 amino acids (exons 1–3) [23,58]. Further splicing generates full-length IE72 and IE86 mRNA containing transcripts from exons 4 and 5, respectively (Figure 1A, upper panel) [23,59,60,61,62]. The initial production of IE72 is a critical process in primary HCMV infection and reactivation from latency [13,15,63,64,65,66]. Therefore, we evaluated several miRNAs that potentially impact IE72 synthesis by identifying candidate molecules that bound to the 3′-UTR of *UL123* by subjecting argonaute-crosslinking immunoprecipitation (AGO-CLIP)-seq and mRNA-seq data to bioinformatic prediction [27]. This indicated that miR-200b-3p and miR-200c-3p play principal roles in host protective mechanisms during lytic replication [23,34,35,40,41,58,59]. We investigated changes in the expression of MIE proteins and HCMV viral load in response to miR-200b-3p and miR-200c-3p upregulation during the early stage of active HCMV replication and latent infection. We also investigated the mechanism by which these two miRNAs modulate the phosphorylation of the RelA/p65 subunit [67,68,69]. 

## 2. Results

### 2.1. Identification of miRNAs That Bind to the 3′-UTR of UL123 Based on AGO-CLIP-Seq Data

We searched for hsa-miRs that could bind to the 3′-UTR of *UL123* using bioinformatic analysis. We obtained raw high-throughput AGO-CLIP-seq and mRNA-seq datasets from cultured human fibroblasts infected with the Towne strain of HCMV representing acute lytic infection (GEO accession number: GSE63797) [27]. We compared enriched regions of AGO-CLIP-seq with background mRNA-seq reads with a total of 406 base pairs (bp) within the 3′-UTR of *UL123* 2 days post-infection (dpi) (maximal reads: 4059 vs. 1591, respectively; Figure 1A, lower panel). The height of the mapped reads was similar between AGO-CLIP-seq and mRNA-seq at 1 (1011 vs. 1282) or 3 dpi (2713 vs. 1444) (Appendix A). Thereafter, eight miRNAs with the most normalized transcripts per million (TPM) at 2 dpi were selected as having a high likelihood of binding to the 3′-UTR of *UL123*. These miRNAs harbored canonical binding sites (8mer, 7mer-A1, and 7mer-m8) that match the seed region. miR-1208/-4676-3p/-892c-3p or -662/-4716-5p had identical seed sequences (Figure 1A,B).

### 2.2. Binding of the Predicted miRNAs to the 3′-UTR of UL123

We evaluated the binding of miRNAs to the 3′-UTR of mRNA by using an assay that employed the intracellular expression of green fluorescent protein (GFP) as an indicator of binding. The GFP-based cell assay allows diverse analyses of GFP expression via immunoblotting, immunofluorescence, and flow cytometry. It also enables more reproducible verification of the degrees of miRNA binding through the permanent and stable incorporation of EGFP plasmids harboring the DNA fragments of interest into cells, which facilitates GFP-positive cell sorting [70,71]. To ensure better results, we performed the GFP assay in human glioblastoma astrocytoma (U373MG) cells that exhibit a high transfection efficiency for large plasmids (pEGFP-N1 vector: 4700 bp) and small RNAs [72,73]. Flow cytometry and fluorescence microscopy revealed that U373MG cells stably transfected with EGFP plasmids (SCs) expressed GFP (Figure 1C,D). The percentage of GFP-positive cells in *UL123*-3′-UTR-SC (SCs transfected with oligonucleotides corresponding to the 3′-UTR of *UL123* mRNA) was as high as that in positive control cells transfected with an empty EGFP plasmid (pMOCK), even when the transfection rate of *UL122-123*-3′-UTR-SCs (transfected with oligonucleotides encoding the 3′-UTRs of *UL122* and *UL123* mRNAs) was 45% (Figure 1C,D).

In our in-house EGFP-based assay, reduced GFP expression was observed when the miRNA bound to the 3′-UTR of mRNA in the SCs. Eight miRNAs were predicted based on the results of AGO-CLIP-seq (a negative control (NC) mimic was included in vitro to ensure the absence of off-target effects) [74]. These miRNAs were transiently transfected into *UL123*- or *UL122-123*-3′-UTR-SCs for 2 days during the same period [27]. *UL123*- and *UL122-123*-3′-UTR-SCs transiently transfected with the eight miRNAs expressed GFP (Figure 1E,F). The number of GFP-positive cells normalized to that in cells transfected with the NC mimic for all eight miRNAs did not differ among pMOCK, *UL123*-, and *UL122-123*-3′-UTR-SCs (Figure 1E,F). These findings indicated that the miRNAs identified by AGO-CLIP-seq do not bind to the 3′-UTR of *UL123.*

### 2.3. MiR-200b-3p and miR-200c-3p Bound to the 3-UTR of HCMV UL123

We found that miR-200b-3p and miR-200c-3p had consistently high normalized TPM values (medians of three dpi; 17.7 and 17.9, respectively) for the 3′-UTR of *UL123* at 1, 2, and 3 dpi in the AGO-CLIP-seq and mRNA-seq datasets [27]. Using the BiBiServ RNA hybrid program, we predicted that miR-200b-3p and miR-200c-3p would bind to the 3′-UTR of *UL123* with non-canonical 5-mer matching in low free energy (median of three dpi; −18.7 and −20.6 kcal/mol, respectively; Appendix A). Based on this observation and recent findings regarding the essential role played by non-canonical 5-mer binding such as 5mer-m2.6 or 5mer-A1 in hsa-miRs [75,76,77,78,79,80], we investigated whether miR-200b-3p and miR-200c-3p can bind to the 3′-UTR of *UL123* and *UL122* and whether they can affect IE72 expression and HCMV replication.

We generated SCs using an EGPF plasmid harboring mutated oligonucleotides corresponding to the predicted miR-200b-3p- and miR-200c-3p binding sequences within the 3′-UTR of *UL123* (Δ*UL123*-3′-UTR SCs) (Table 1). The Δ*UL123*-3′-UTR SCs that were not transiently transfected with miRNA mimics or infected with HCMV also expressed GFP (Figure 1C,D). Then, we transiently transfected Δ*UL123*-, *UL123*-, or *UL122-123*-3′-UTR-SCs with miRNA-200b-3p or miR-200c-3p mimics for 2 days. Exogenously provided miR-200b-3p and miR-200c-3p significantly decreased the numbers of GFP-positive *UL123*- and *UL122-123*-3′-UTR SCs (all *p* < 0.001) after normalization to those in NC cells transfected with the mimic (Figure 2A,B). However, treatment with exogenous miR-200b-3p or miR-200c-3p in the Δ*UL123*-3′-UTR SCs or pMOCK did not affect GFP-positive cells (Figure 2A,B). 

Immunoblotting findings also revealed that exogenous miR-200b-3p and miR-200c-3p reduced GFP expression compared with the transfected NC mimic in *UL123*-3′-UTR and *UL122-123*-3′-UTR SCs (Figure 2C). The GFP/β-actin values were normalized to cells that were not transfected with the mimic (all *p* < 0.001; Figure 2D). However, GFP levels in the Δ*UL123*-3′-UTR SC and pMOCK did not differ among cells transfected with NC mimic, miR-200b-3p mimic, or miR-200c-3p mimic (Figure 2C,D). The expression of GFP was similar to that determined by flow cytometry and immunoblotting (Figure 2E). The GFP fluorescence was less intense in *UL123*- (Figure 2E, panels k and l) and *UL122-123*-3′-UTR SCs (Figure 2E, panels o and p) transfected with miR-200b-3p or miR-200c-3p mimic compared with those in mutant *UL123*-3′-UTR SCs (Figure 2E, panel g and h). Overall, miR-200b-3p and miR-200c-3p bound to the 3′-UTR of HCMV *UL123*, and the 3′-UTR of *UL122* had ≈50% of the reported binding affinity [34,35].

### 2.4. Restoration of Downregulated Expression of miR-200b-3p and miR-200c-3p after Acute Lytic HCMV Infection Decreased IE72 and IE86 Expression and HCMV Titers

We assessed the levels of endogenous miRNA after HCMV infection and the downstream effects of the binding of miR-200b-3p and miR-200c-3p to the 3′-UTR of *UL122* and *UL123* [34,35] in the human foreskin fibroblast-1 (HFF-1) cell line that is highly permissive for acute lytic replication. We infected these cells with HCMV Towne strain at a multiplicity of infection (MOI) of 0.1 [27,81,82,83,84,85].

Such infection resulted in the continuous suppression of miR-200b-3p and miR-200c-3p until 7 dpi in contrast to uninfected cells that were not affected (Figure 3A). MiR-200b-3p and miR-200c-3p were reduced to similar ratios in HFF-1 cells with HCMV infection (means ± standard deviation, 0.35 ± 0.03 vs. 0.36 ± 0.02, *p* = 0.887; Figure 3B, left panel, data are from nine measurements each on days 1, 2, 3, 5, and 7). The suppressed ratio in miR-200b-3p significantly correlated with the ratio in miR-200c-3p (*r* = 0.89, *p* < 0.001; Figure 3B, right panel, data from nine measurements each on days 1, 2, 3, 5, and 7).

Abundant IE72 and IE86 are generally synthesized at 2–3 dpi in permissive cells infected with a laboratory-adapted HCMV strain [27,83,85,86]. We also found abundant IE72 and IE86 expression at 2 dpi in the nuclei of HFF-1 cells infected with the HCMV Towne strain at an MOI of 0.1 (Appendix A). According to these findings, we measured IE72 and IE86 levels as well as HCMV titers at 2 dpi after transient transfection with miR-200b-3p or miR-200c-3p mimics under the same conditions in vitro [27]. 

Restoration of the repressed endogenous miR-200b-3p and miR-200c-3p after acute HCMV infection via mimic transfection resulted in significantly decreased IE72 and IE86 expression (all *p* < 0.001) compared with the immunoblots of HFF-1 cells transfected with NC mimic (Figure 3C,D). To confirm the effect of miR-200b-3p and 200c-3p on IE72 and IE86 production, we transfected the specific antisense inhibitor of each microRNA after 1 h in HFF-1 cells simultaneously infected with HCMV Towne strain and transfected with miR-200b-3p or miR-200c-3p mimics. Treatment with a miRNA inhibitor, which can downregulate the expression miR-200b-3p and miR-200c-3p in the background of exogenous microRNA mimic exposure during HCMV lytic replication, resulted in the expression of IE72 and IE86 to levels similar to those observed in HFF-1 cells infected with HCMV and transfected with the NC mimic (Figure 3E). 

Since the major IE proteins play essential roles in continuous lytic replication and progeny HCMV production [5,13,14,15,16,17,18,21,22,23], miR-200b-3p and miR-200c-3p, which suppress IE72 and IE86 synthesis via post-transcriptional modification, should restrict active HCMV replication. We found that miR-200b-3p and miR-200c-3p mimic significantly reduced HCMV titers in cell lysates (−log_10_ (4.1 ± 1.5) and −log_10_ (4.4 ± 1.3) IU/µL) and in supernatants (−log_10_ (3.5 ± 1.2) and −log_10_ (3.5 ± 1.6) IU/µL) (three independent measurements each with three replicates of lysates or supernatants each at 1, 2, 3, 5, and 7 dpi) compared with NC mimic-transfected cells at 1, 2, 3, 5, and 7 dpi (all *p* < 0.001) (Figure 3F). These results showed that miR-200b-3p and miR-200c-3p directly downregulate the expression of IE72 and IE86, which is a phenomenon that results in inhibited HCMV replication.

The expression of HCMV *IE72* and *IE86* mRNAs can start 4–6 h post-infection (hpi) regardless of the differences in cell types or the size of the viral inoculum [87,88]. Decrease in the expression of miR-200b-3p and miR-200c-3p started at 2 hpi in acute lytic-infected cells, which preceded IE72 and IE86 expression (Figure 4). This finding also showed that immediately after acute infection, HCMV induced the downregulation of miR-200b-3p and miR-200c-3p to maintain the production of lytic proteins.

### 2.5. The Expression of miR-200b-3p and miR-200c-3p Is Not Downregulated during Latent HCMV Infection and Exogenous Inhibition of miRNA Expression Induced IE Expression and Virus Replication during Latency

We evaluated whether miR-200b-3p and miR-200c-3p levels are altered and similarly lead to suppressed IE gene expression during latent HCMV infection, as it is under acute lytic replication. The non-permissive human monocytic leukemia (THP-1) cell line, which is a suitable model of latent HCMV infection, was infected with the HCMV Toledo strain at an MOI of 5 [28,89,90,91]. Previous studies detected *UL123* mRNA at 1, 2, and 3 dpi, but not at 7 and 10 dpi in THP-1 cells infected with the Toledo strain at a high MOI [28,91]. 

We also verified early UL122 and UL123 mRNA expression in THP-1 cells infected with the Toledo strain without mimic transfection but not in latent-infected THP-1 after 7 dpi (Appendix A) [28]. Endogenous miR-200b-3p and miR-200c-3p levels were significantly decreased at 1, 2, and 3 dpi in THP-1 cells with HCMV infection and without mimic transfection compared with THP-1 cells without HCMV infection. These results were consistent with those of acute lytic infection. However, the levels of the two miRNAs did not differ between THP-1 cells with and without HCMV infection at 7, 10, and 14 dpi (Figure 5A). High-throughput screening of cellular miRNAs in latent-infected THP-1 cells at 10 dpi has not identified any changes in the levels of miR-200b-3p or miR-200c-3p [28]. These findings indicated that downregulated miR-200b-3p and miR-200c-3p when IE proteins are expressed return to levels equivalent to those of cells without lytic HCMV replication during latency.

Based on this finding, we transfected THP-1 cells with miR-200b-3p or miR-200c-3p inhibitors immediately after HCMV infection to downregulate the two miRNAs during quiescence; then, we measured IE72 and IE86 levels and HCMV titers at various time points until 14 dpi. Both IE72 and IE86 were undetectable in quiescent, latent-infected THP-1 cells (7, 10, and 14 dpi in Figure 5B). However, initial transfection with inhibitors resulted in abundant expression of IE72 and IE86 at 7, 10, and 14 dpi compared with that in HCMV-infected THP-1 cells without inhibitor transfection at 10 dpi (Figure 5C). HCMV titers after 7 dpi were significantly increased in THP-1 cells that were initially transfected with inhibitors in contrast to those without inhibitor or NC mimic transfection (Figure 5D). These findings indicated that miR-200b-3p and miR-200c-3p can regulate the shift from latent-to-lytic phase transition of HCMV. 

### 2.6. IE Proteins and MiR-200b-3p or miR-200c-3p Are Associated with Phosphorylation of RelA/p65 at Ser536

The NF-κB binding to MIEP can potentially and rapidly induce the activation of genes encoding the IE72 and IE86 during acute HCMV lytic infection. Therefore, there could be a substantial relationship between the activation of NF-κB and the production of IE proteins. We confirmed that TNF-α appropriately induced RelA/p65 phosphorylation at Ser^536^ (p-Ser^536^ RelA/p65) within 1 h, which was consistent with previous findings (Appendix A) [52,53,55,56]. However, p-Ser^536^ RelA/p65 levels were significantly low from 6 to 24 hpi even though the expression of RelA/p65 was maintained in HFF-1 cells infected with the HCMV Towne strain at the same time points (Figure 6A).

MiR-200b-3p and miR-200c-3p affected IE72 and IE86 expression after binding to the 3′-UTR of *UL122* and *UL123* and were downregulated during acute lytic replication, and the levels of p-Ser^536^ RelA/p65 continuously decreased during the extensive synthesis of IE72 and IE86. We speculated that IE proteins are associated with p-Ser^536^ RelA/p65 or that p-Ser^536^ RelA/p65 interacts with the HCMV MIEP, which is a major transcriptional promotor of IE protein production [13,14,43]. We transfected U373MG cells with plasmids encoding IE72 and IE86 and measured levels of RelA/p65 and p-Ser^536^ RelA/p65 after 2 days. U373MG cells expressing IE72 and IE86 protein did not express p-Ser^536^RelA/p65, even though RelA/p65 was expressed in U373MG cells transfected with IE72 or IE86 plasmids (Figure 6B). These initial findings indicated that IE72 and IE86 inhibit the phosphorylation of RelA/p65 at Ser^536^ and that acute infection with HCMV does not induce RelA/p65 phosphorylation at Ser^536^ (unlike the case observed in the background of infection with other microbes) [52,53].

We investigated whether recovery of the downregulated endogenous miR-200b-3p and miR-200c-3p, which can decrease IE72 and IE86 protein, changes p-Ser^536^ RelA/p65 during acute lytic HCMV replication. Numerous miR-200b-3p- or miR-200c-3p-mimic-transfected and HCMV-infected cells exhibited high levels of p-Ser^536^ RelA/p65 at 2 dpi; these levels were as high as those observed in RelA/p65-positive cells (Figure 6C,D). However, significantly fewer cells were p-Ser^536^ RelA/p65-positive among those transfected with NC mimic (all *p* < 0.001, Figure 6C,D). In contrast, further downregulation of miR-200b-3p and miR-200c-3p using inhibitors resulted in a significant reduction in the levels of nuclear and cytoplasmic p-Ser^536^ RelA/p65 (Figure 6E, panels l or o) compared with that in cells transfected with an miR-200b-3p mimic (Figure 6E, panel f) or miR-200c-3p mimic (Figure 6E, panel i). Towne strain-infected HFF-1 cells transfected with miR-200b-3p and miR-200c-3p mimics expressed significantly more p-Ser^536^ RelA/p65 than HFF-1 cells without mimics or with NC mimic transfection but did not alter the levels of RelA/p65 or p105/p50 protein (Figure 6F,H-left panel). Restoring the levels of endogenous miR-200b-3p and miR-200c-3p through inhibitor transfection 1 h after simultaneous HCMV infection and mimic transfection decreased p-Ser^536^ RelA/p65 expression to the minimal levels (Figure 6G,H-right panel). These findings indicated that abundant miR-200b-3p or miR-200c-3p expression leads to post-transcriptional inhibition of IE72 and IE86 expression and that this is associated with the induction of p-Ser^536^ RelA/p65 expression. 

### 2.7. Phosphorylated RelA/p65 (Ser^536^) Binds to MIEP Region

We used cross-linking chromatin immunoprecipitation (ChIP) assays to investigate whether RelA/p65 phosphorylated at Ser^536^ interacts with the enhancer region containing NF-κB-binding sites in MIEP to uncover how the expression of IE proteins is correlated with the reduction in p-Ser^536^ RelA/p65 levels. We found that the RelA/p65 and p-Ser^536^ RelA/p65 bound to the HCMV MIEP within 1 h of Towne strain infection (Figure 7A). The relative amount of RelA/p65 bound to MIEP was continuously increased compared with that of p-Ser^536^ RelA/p65 (Figure 7A). The ratio of RelA/p65 to p-Ser^536^ RelA/p65 increased over time (Figure 7C).

In contrast to natural acute HCMV infection, transfection with miR-200b-3p and miR-200c-3p mimics induced the binding of p-Ser^536^ RelA/p65 to MIEP to a greater extent than that observed with RelA/p65 (Figure 7B). The ratio of RelA/p65 to p-Ser^536^ RelA/p65 at MIEP decreased over time in HFF-1 cells transfected with miR-200b-3p/200c-3p mimics with significant changes at 6 and 24 hpi for both miR-200b-3p and miR-200c-3p. Opposite results were observed in cells without mimic transfection (Figure 7C). We also found that the upregulation of miR-200b-3p and miR-200c-3p results in decreased levels of pro-inflammatory interleukin (IL)-1β, IL-6, and RANTES (CCL5), all of which are induced in response to NF-κB activation (Figure 7D). These data suggested that downregulating the expression of IE proteins by upregulating the levels of miR-200b-3p and miR-200c-3p can result in the induction of p-Ser^536^ RelA/p65 expression. Thereafter, instead of RelA/p65, p-Ser^536^ RelA/p65 binds to the MIEP owing to its abundance and finally results in the inhibition of NF-κB activity and pro-inflammatory cytokine production.

## 3. Discussion

The discovery of new biological agents targeting IE protein synthesis during acute HCMV replication and immediately after the lytic switch from latent infection is essential to improve early detection, prognostic prediction, design preventive strategies, and ultimately control the harmful effects from HCMV in the general population as well as critically ill or severe immunocompromised patients [13,15,34,92]. MicroRNAs and short hairpin (sh) RNAs are promising candidates for silencing gene expression via post-transcriptional regulation [24,29,31,35,66,93]. The exploration of miRNAs generally requires the prediction of the seed sequences that bind to the 3′-UTR using various algorithms and de novo or machine learning tools, but various computational analyses could not identify enough binding ability or downstream biological roles of the predicted miRNAs in vitro or in vivo. Accordingly, we selected new potential miRNAs that can bind to the 3′-UTR of HCMV *UL123*, which downregulates IE72 expression, using high-throughput sequencing data from AGO-CLIP-seq instead of traditional algorithms in silico, because cellular miRNAs interacting with *UL123*-3′-UTR have not been evaluated despite the crucial role of IE72 during early infection [13,27,59,63,87,94,95]. However, sophisticated SC-based flow cytometry of GFP expression revealed that miRNAs with the most abundant transcripts in AGO-CLIP-seq from HCMV-infected cells did not bind to the 3′-UTR of *UL123*.

Thus, the inability of high-throughput sequencing to predict the roles of miRNAs was verified, and new high AGO-CLIP-seq transcripts for miR-200b-3p and miR-200c-3p that are capable of binding to the 3′-UTR of *UL123* were identified. An association between miR-200b-3p and miR-200c-3p and IE86 suppression through binding to the 3′-UTR of *UL122* supported the hypothesis that miR-200b-3p and/or 200c-3p function during the IE stage of HCMV replication by targeting the 3′-UTR of *UL123* and are associated with IE72 expression [34,35,40]. The transient transfection of SCs, flow cytometric measurements of GFP, immunoblotting, and fluorescence imaging showed that miR-200b-3p and miR-200c-3p bind to the 3′-UTR of *UL123*. The binding affinity of miR-200b-3p and miR-200c-3p for the 3′-UTR of *UL122* and *UL123* was consistently similar. We and others have also found that fluorescence emission is reduced by 50% in luciferase reporter assays with the MIEP promoter [34,35]. These findings provide tangible evidence of the interactions between miR-200b-3p as well as miR-200c-3p and the 3′-UTR of *UL122-123* (exons 4 and 5). 

The results of the reporter assay using the GFP-expressing SC system correlated with a decrease in IE72 and IE86 protein levels as well as HCMV titers after treatment with miR-200b-3p and miR-200c-3p. In addition, miR-200b-3p and miR-200c-3p inhibitors that counterbalanced the effects of exogenously supplied miRNA returned the expression of IE72 and IE86 to initial levels. These data support the hypothesis that upregulated miR-200b-3p and miR-200c-3p downregulate the expression of IE72 and IE86 after binding to the 3′-UTR of *UL122-123* and finally inhibit HCMV replication. In contrast, HCMV can trigger the downregulation of miR-200b-3p and miR-200c-3p to facilitate the expression of IE72 and IE86 during early acute infection. The levels of miR-200b-3p or miR-200c-3p were inversely correlated with the expression of IE72 and IE86 and HCMV replication in latent-infected THP-1 cells. 

The site-specific post-translational phosphorylation of RelA/p65, RelB, and c-Rel subunits after the degradation of inhibitor of NF-κB (IκBα) in the canonical NF-κB pathways plays an important role in modulating the downstream effects of NF-κB in response to a wide range of inflammatory stimuli [67,68,96]. Researchers have found a connection between IE proteins and MIEP and the canonical NF-κB signaling cascade, and they identified that this phenomenon involves the regulation of the activities of the IKK complex (IκB kinase) or modulation of NF-κB activity [49,97,98]. The I kappa kinases IKKα, IKKβ, or IKKε also phosphorylate the Ser^468^ or Ser^536^ residues in the RelA/p65 subunit of NF-κB, which is an event that results in an improved ability of NF-κB to bind the DNA in response to inflammatory stimuli [52,53,68,99,100]. Therefore, phosphorylation at the Ser^536^ residue in the transactivation domain of RelA/p65 can facilitate the transcriptional activity of NF-κB [52,53,54,57,101,102]. However, others have found an antagonistic role of IKK-mediated phosphorylation of Ser^536^; they indicate that this even contributes to the suppression of NF-κB activity by accelerating the turnover of the RelA/p65 or promoting the proteosomal degradation of the RelA/p65 [103,104,105]. The negative regulation of NF-κB by p-Ser^536^ RelA/p65 might protect from exaggerated inflammation and a detrimental cytokine storm caused by NF-κB hyperactivation [103,105,106]. 

This study on acute HCMV infection showed that the p-Ser^536^ RelA/p65 negatively affects IE protein levels and NF-κB activity by binding to MIEP. The expression of IE proteins is associated with inhibition of the phosphorylation of RelA/p65 at Ser^536^, and the levels of p-Ser^536^ RelA/p65 are decreased during acute lytic infection of HCMV when IE proteins are expressed. Contrary to these findings, the upregulation of miR-200b-3p and miR-200c-3p which helps downregulate the expression of IE72 and IE86 resulted in increased p-Ser^536^ RelA/p65 expression. This could competitively interrupt the binding of NF-κB to MIEP and then inhibit the production of pro-inflammatory cytokines. Therefore, we propose that increasing the levels of miR-200b-3p and miR-200c-3p can result in the regulation of lytic replication of HCMV (Figure 8).

Our findings are supported by the published results and methodology. The correlation between the expression of miR-200b-3p and miR-200c-3p was constant (miR-200c-3p > miR-200b-3p) [35,40], and there was a close correlation between the downregulated levels of miR-200b-3p and miR-200c-3p. The SC-based system of using vectors expressing GFP to score for the 3′-UTR binding activity of miRNA was sensitive. The numbers were sufficient to provide statistical power and reproducibility. We attenuated the effects of miR-200b-3p and miR-200c-3p using antisense inhibitors.

Our experiments have the following limitation. There would be conceptual debates as to whether THP-1 cells represent a true HCMV latency model, even though many studies used THP-1 cells for an experimental latent-reactivation model [28,89,91,107,108]. It would be quite informative to evaluate the effect of miR-200b-3p/-200c-3p and their regulation in other models of latency as well as in normal human bone marrow cells (or more specifically CD34^+^ HPCs). If we could confirm the role of two microRNAs on HCMV lytic and latent infection from various human cells, we will be able to secure more evidence to apply the research results to humans.

In summary, the following findings of this study were novel. MiR-200b-3p and miR-200c-3p inhibited IE72 production after binding to the 3′-UTR of *UL123* and decreased the HCMV titers. The downregulation of miR-200b-3p and miR-200c-3p can result in IE72 and IE86 production in the latent-infected cells. The levels of phosphorylated Ser^536^ RelA/p65 are decreased during lytic replication, and IE proteins alone can inhibit the phosphorylation of p-Ser^536^ RelA/p65. Phosphorylated RelA/p65(Ser^536^) binds to MIEP. Inhibiting the expression of IE proteins by upregulating the expression of miR-200b-3p and miR-200c-3p results in increased levels of p-Ser^536^ RelA/p65 and prevents hyperinflammation. In conclusion, attenuation of the expression of IE72 and IE86 proteins and decreased HCMV titers in response to the binding of miR-200b-3p and miR-200c-3p to the 3′-UTR of *UL122* and *UL123* is associated with the binding of p-RelA/p65 Ser^536^ to MIEP. Accumulating information about the functions and effects of miR-200b-3p and miR-200-3p might support a potential clinical role for the use of these miRNAs as biomarkers or therapeutic targets and contribute to our understanding of the role of hsa-miRs during the immediate early period of HCMV lytic and latent infection. 

## 4. Materials and Methods

### 4.1. Cell Culture and HCMV Infection

We obtained materials from the listed suppliers as follows: HFF-1 and THP-1 cell lines (ATCC, Manassas, VA, USA); U373MG cell lines (Korean Cell Line Bank, Seoul, Korea). HFF-1 cells were cultured in Dulbecco’s modified Eagle’s medium (Thermo Fisher Scientific Inc., Waltham, MA, USA Fisher, Seoul, Korea) supplemented with 15% heat-inactivated fetal bovine serum and L-glutamine (both from Thermo Fisher Scientific Inc.). THP-1 and U373MG were cultured in RPMI-1640 (Welgene Inc., Gyeongsangbuk-do, Korea) containing 10% FBS and L-glutamine. All cell cultures were maintained with penicillin (50 U/mL) and streptomycin (50 μg/mL) in a humidified 5% CO_2_ atmosphere at 37 °C. The HFF-1 and U373MG cells were infected with HCMV Towne strain (ATCC, Manassas, VA, USA) at an MOI of 0.1 to establish acute lytic replication [27,81,82,83,84]. We latently infected THP-1 cells with the HCMV Toledo strain (low-passage clinical isolate) at an MOI of 5 [28,89,90]. Cells and culture supernatants were harvested at the indicated hours (hpi) or days (dpi) post-infection.

### 4.2. Construction of IE Plasmids and Plasmids Containing the 3′-UTR of UL123

Oligonucleotides corresponding to the 3′-UTR of *UL123* (wild type, 112 bp and mutant, 112 bp) (Table 1), containing restriction sites for XhoI and EcoRI (Takara Bio, Mountain View, CA, USA), were cloned into pEGFP-N1 plasmids (4700 bp; Takara Bio, Mountain View, CA, USA) using a standard procedure. Another plasmid harboring oligonucleotides (232 bp, Table 1) corresponding to the 3′-UTR of *UL122* and *UL123* (*UL122-123*-3′-UTR-EGFP) was constructed by digesting the oligonucleotides corresponding to the 3′-UTR of *UL122* (102 bp) using EcoRI and then ligating the digested oligonucleotides downstream of *UL123*-3′-UTR-EGFP. All pEGFP plasmids harboring the aforementioned constructs were linearized using BsaI for transfection into U373MG cells. Plasmids expressing wild-type IE72 and IE86 were generated as described [44,109,110,111]. Identical oligonucleotide inserts were verified by Sanger sequencing in all plasmids.

### 4.3. Generation of Cell Lines Stably Transfected with 3′-UTR-EGFP

U373MG cell lines were transfected with linearized WT *UL123*-, mutant *UL123*-, and *UL122-123*-3′-UTR-pEGFP, and empty pEGFP (pMOCK, positive control) for 5 days using Lipofectamine^®^ LTX & PLUS^TM^ (Invitrogen, Carlsbad, CA, USA). Cells were incubated in Trypsin-EDTA (Welgene., Gyeongsangbuk-do, Korea) and washed with PBS. After resuspension and filtration through 70-micrometer strainers, cells were sorted using a FACSAria^TM^ (BD Biosciences, San Jose, CA, USA). Ten thousand GFP-positive cells were collected from each transfected group, and three types of 3′-UTR-pEGFP- or pMOCK-SC were maintained in RPMI-1640 supplemented with 10% FBS. 

### 4.4. Measurement of the Binding of miRNAs to the 3′-UTR Targets

MiRNA mimics (300 ng/well) for *UL123* and miR-200b-3p/miR-200c-3p were transiently transfected in U373MG 3′-UTR-SC (1 × 10^6^/well) using Lipofectamine^®^ RNAiMAX (Invitrogen, Carlsbad, CA, USA) in 6-well plates. GFP-positive cells and levels of GFP expression were quantified using flow cytometry (BD FACSCanto II^TM^, BD Biosciences, San Jose, CA, USA) and immunoblotting, respectively, 2 days after transfection. 

### 4.5. Evaluation of MicroRNA Expression and HCMV Viral Loads

Total small RNA (50 ng) was purified using mirVana^TM^ miRNA isolation kits (Invitrogen, Carlsbad, CA, USA) and reverse-transcribed in a thermal cycler using TaqMan^TM^ MicroRNA Reverse Transcription Kits (Applied Biosystems, Foster City, CA, USA). Complementary DNA was amplified by real-time polymerase chain reaction (PCR) using stem-loop primers (Appendix A) designed for specific miRNAs (TaqMan^TM^ MicroRNA Assays, Applied Biosystems, Foster City, CA, USA) and a LightCycler^®^ 480 (Roche Life Science, Penzberg, Germany) under the following conditions 95 °C for 10 min followed by 45 cycles of 15 sec at 95 °C and 60 sec at 60 °C. For absolute quantitation, seven-point standard curves of serial dilutions (from 10^1^ to 10^7^) were prepared from a mirVana^TM^ miRNA mimic with known copy numbers of miR-200b-3p/200c-3p (Invitrogen, Carlsbad, CA, USA). HCMV titers (IU/mL) were measured by performing quantitative real-time PCR for *UL83* using the LightCycler^®^ 480 (Table 1). Standard curves were constructed from serial dilutions of the National Institute for Biological Standards and Control 09/162, which is the first WHO international standard of HCMV for nucleic acid amplification [112]. All quantitative real-time PCR results are expressed as copy numbers/µL of total input RNA or DNA mass.

### 4.6. Immunoblotting

Transfected cells homogenized in radioimmunoprecipitation assay buffer were centrifuged; then, total protein (20 μg) was separated by sodium dodecyl sulfate–polyacrylamide gel electrophoresis and transferred to polyvinylidene difluoride membranes (Sigma-Aldrich, St. Louis, MO, USA). Non-specific protein binding was blocked by 5% skim milk in 0.05% Tween-20 in Tris-buffer at 37 °C for 1 h; then, the membranes were incubated overnight at 4 °C with mouse anti-GFP (Santa Cruz Biotechnology, Inc., Dallas, TX, USA), mouse anti-HCMV IE72 and mouse anti-HCMV IE86 (both from Santa Cruz., Dallas, TX, USA), rabbit anti-RelA/p65, rabbit anti-phospho-RelA/p65 (Ser^536^), and rabbit anti-p105/p50 (all from Cell Signaling Technology, Danvers, MA, USA), or mouse anti-ß-actin (Sigma-Aldrich) monoclonal antibodies (Appendix A). Then, the membranes were incubated with horseradish peroxidase-conjugated IgG for 1 h, and proteins were visualized using a chemiluminescent substrate. Band density was quantified using Image J software (http://rsbweb.nih.gov/ij/, accessed on 8 December 2021) and normalized to the level of β-actin. 

### 4.7. Flow Cytometry 

Cells were washed and stained with mouse anti-GFP-Alexa Fluor^®^ 488 conjugate (Santa Cruz., Dallas, TX, USA), rabbit anti-RelA/p65-Alexa Fluor^®^ 488 conjugate, rabbit anti-phospho-RelA/p65 (Ser^536^)-Alexa Fluor^®^ 488 conjugate, or rabbit anti-IgG-isotype control-Alexa Fluor^®^ 488 conjugate (all from Cell Signaling, Danvers, MA, USA) as a negative control for 1 h (Appendix A). Cells were stained with fluorescein isothiocyanate (FITC) anti-human IL-1β and FITC anti-human IL-6 (both from Invitrogen), or FITC anti-human RANTES (CCL5) (Novus Biologicals, Centennial, Colorado, CO, USA) to investigate the expression of intracellular cytokines (Appendix A). Fluorescence intensity of 10,000 cells at 488 nm was measured using BD FACSCanto II. Permeabilization and Fcγ receptors were blocked using 5% horse serum (Gibco) to stain intracellular proteins.

### 4.8. Immunocytochemistry

We cultured HFF-1 cells (1 × 10^6^/well) on sterile gelatin-coated coverslips in 6-well plates. Permeabilization and Fcγ receptors were blocked; then, the cells were stained with mouse anti-HCMV IE72, mouse anti-HCMV IE86, or rabbit anti-phospho-RelA/p65 (Ser^536^)-Alexa Fluor^®^ 488 conjugate overnight at 4 °C, which was followed by incubation with goat anti-mouse IgG secondary antibody for HCMV IE72 (FITC-conjugated) or IE86 (rhodamine-conjugated) (Invitrogen, Carlsbad, CA, USA) for 1 h (Appendix A). Fluorescence images were obtained using an Axio Observer Z1 LSM 780 confocal microscope (Carl Zeiss AG, Oberkochen, Germany) with a photomultiplier tube for light transmission (Carl Zeiss AG, Oberkochen, Germany). 

### 4.9. Cross-Linking Chromatin Immunoprecipitation (ChiP)

We infected HFF-1 cells (1 × 10^6^/well) with HCMV Towne strain, incubated them with formaldehyde for 15 min, and then quenched cross-linking with glycine. The cross-linked HFF-1 cells were harvested and lysed in immunoprecipitation (IP) buffer containing protease inhibitors. After washing and resuspension, cell pellets were sonicated in IP buffer (Appendixes, Appendix A) and then incubated with rabbit anti-RelA/p65 (Cell Signaling, Danvers, MA, USA) or mouse anti-phospho-RelA/p65 (Ser^536^) (Santa Cruz., Dallas, TX, USA) in an ultrasonic water bath for 15 min at 4 °C. Protein A-agarose beads were washed; then, IP samples were boiled in Chelex 100 for 10 min and centrifuged. We purified the DNA present in the supernatants after centrifugation and amplified the MIEP containing the NF-κB-binding motifs (proximal 550 bp) using PCR (Table 1) [13,43].

### 4.10. Statistical Analysis

All flow cytometry, immunoblotting, and real-time PCR experiments were conducted in triplicate under the same conditions, and data are shown as means ± standard error of the means (SEM) or means ± standard deviation. Flow cytometry results were analyzed using FlowJo^TM^ (version 10.7, BD Biosciences). All other graphs were constructed using Prism Software version 8.0 (GraphPad Software Inc., San Diego, CA, USA). Two groups were compared using unpaired independent *t*-tests. Values were analyzed using SPSS version 25.0 (IBM Corp., Armonk, NY, USA) and those with *p* < 0.05 (two-tailed) were considered statistically significant.

## 5. Conclusions

The human microRNAs, miR-200b-3p, and miR-200c-3p bind the 3′-UTR of HCMV *UL123* mRNA—that encodes IE72—with non-canonical 5-mer matching. We found that the expression of these two miRNAs at sufficient levels can result in the downregulation of IE72/IE86 and eventually inhibition of HCMV replication during acute lytic and latent infection stages. HCMV by itself can downregulate the expression of miR-200b-3p and miR-200c-3p during lytic replication, and conversely, the downregulated miRNAs can induce IE72/86 expression and virus replication in latent infection. In addition, two miRNAs enhanced the phosphorylation of RelA/p65 (Ser^536^) and decreased the expression of pro-inflammatory cytokines, suggesting a negative regulation of NF-κB activity. These findings highlight the important role played by hsa-miRs during the IE period of HCMV lytic replication and latent infection.

## Figures and Tables

**Figure 1 ijms-23-02769-f001:**
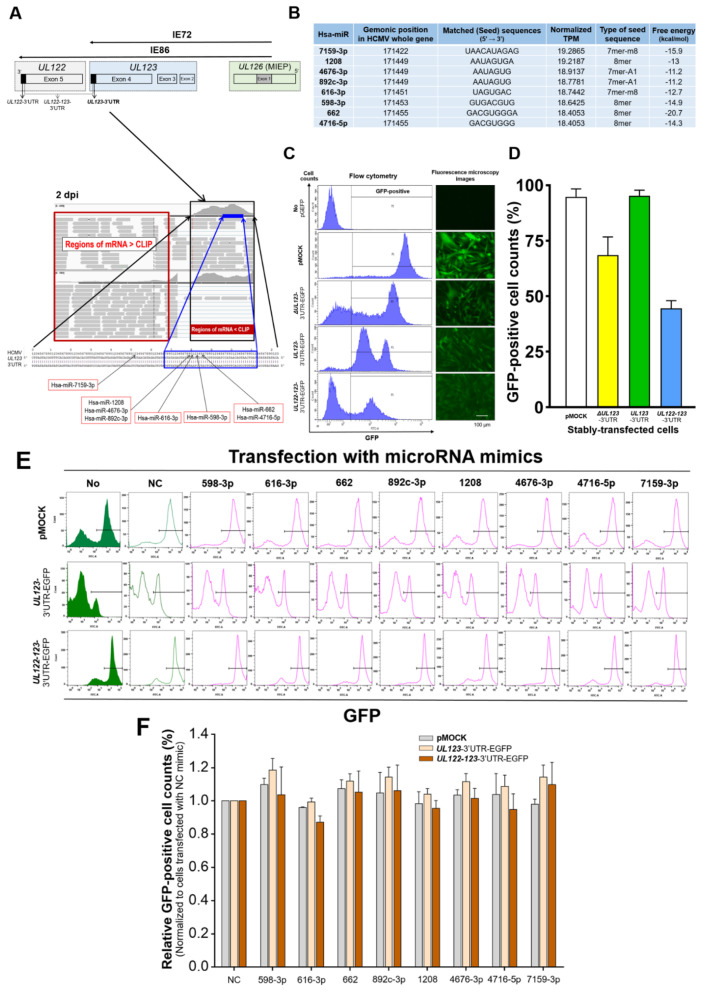
Establishment of U373MG cells stably-transfected with the 3′-UTR and verification of the binding of candidate miRNAs−predicted using AGO-CLIP-seq− to the 3′-UTR of *UL123*. (**A**) Eight candidate hsa-miRs with the most normalized transcripts per million (TPM) at 2 dpi in the 3′-UTR of HCMV *UL123* were selected by subjecting AGO-CLIP-seq and mRNA-seq data to bioinformatic analysis [27]. (**B**) Seed sequence, matched type, and free energy value for the binding of each miRNA to the 3′-UTR of *UL123* were obtained using the BiBiServ RNA hybrid program (https://bibiserv.cebitec.uni-bielefeld.de/rnahybrid, accessed on 17 January 2022). (**C**) Human U373MG cells seeded (1 × 10^6^/well) in 6-well plates were transfected with pEGFP-N1 plasmid vectors harboring mutant *UL123 (*Δ*UL123)*-3′-UTR, *UL123*-3′-UTR, or *UL122-UL123*-3′-UTR for 5 days. GFP-positive cells were sorted from U373MG cells using FACS. We established stably transfected cells (SCs) by incubating sorted cells for 2 weeks in RPMI-1640 medium supplemented with 10% FBS. Positive control U373MG cells were stably transfected with empty pEGFP-N1 plasmid (pMOCK). Cell populations positive for GFP in the SC groups were assayed by flow cytometry and visualized using fluorescence microscopy. (**D**) The transfection efficiency of U373MG cells for EGFP plasmids harboring Δ*UL123*3′-UTR, *UL123*-3′-UTR, or *UL122-UL123*-3′-UTR was determined by counting GFP-positive cells. (**E**) MiRNA mimics (300 ng) for eight candidates that bind to the 3′-UTR of *UL123* and negative control (NC) were transfected into U373MG SCs (pMOCK, *UL122*-3′-UTR-EGFP SC, and *UL122-123*-3′-UTR-EGFP SC; 1 × 10^6^ cells/well) seeded in 6-well plates; exposure was allowed for 2 days in the absence of HCMV infection. The percentage of GFP-positive cells was quantified by flow cytometry, and GFP-positive populations were determined using the FlowJo^TM^ software. (**F**) Relative GFP-positive cell numbers were normalized to those of cells transfected with NC mimics. All quantitative data were generated from triplicate experiments for each miRNA mimic-transfected SC group. Upper lines in box and error bars indicate means and SEM, respectively.

**Figure 2 ijms-23-02769-f002:**
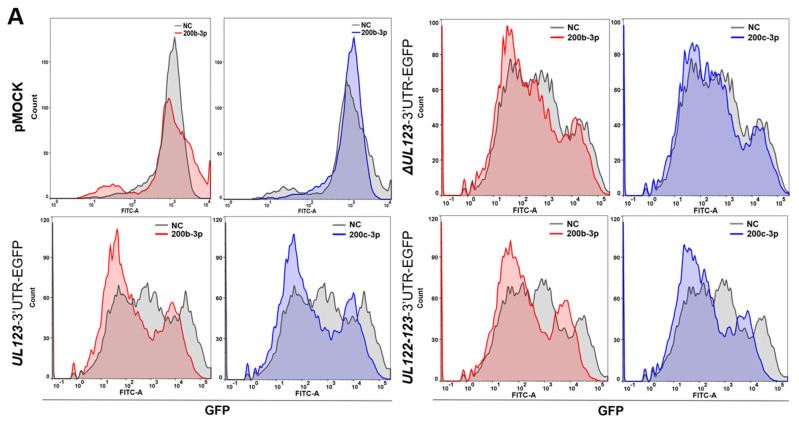
Binding capacity of miR-200b-3p and miR-200c-3p to the 3′-UTR of *UL123* in cells stably expressing transfected GFP. We transiently transfected pMOCK, Δ*UL123*-, *UL123*-, and *UL122-123*-3′-UTR-EGFP U373MG cells (1 × 10^6^/well; 6-well plates) that stably expressed transfected GFP, with miR-200b-3p or miR-200c-3p mimics or NC mimic (300 ng each) for 2 days. Positive control was pMOCK, which indicated that U373MG cells were stably transfected with empty pEGFP-N1. (**A**) Percentage of GFP-positive cells quantified using flow cytometry. Graphs of GFP-positive cells were created using the FlowJo^TM^ software. (**B**) Percentage of GFP-positive cells upon transfection with each miR/NC. (**C**,**E**) Expression of GFP in four EGFP plasmid groups evaluated by immunoblotting and visualized using fluorescence microscopy. (**D**) Relative GFP expression was calculated as GFP/β-actin signal density in each miR-transfected cell (measured using ImageJ) and was expressed as the fold of the value in mimic untransfected cells. All quantitative data represent triplicate experiments involving SCs transfected with miRNA mimics. Upper lines in box and error bars indicate means and SEM, respectively. * *p* < 0.001 vs. NC mimic-transfected cells.

**Figure 3 ijms-23-02769-f003:**
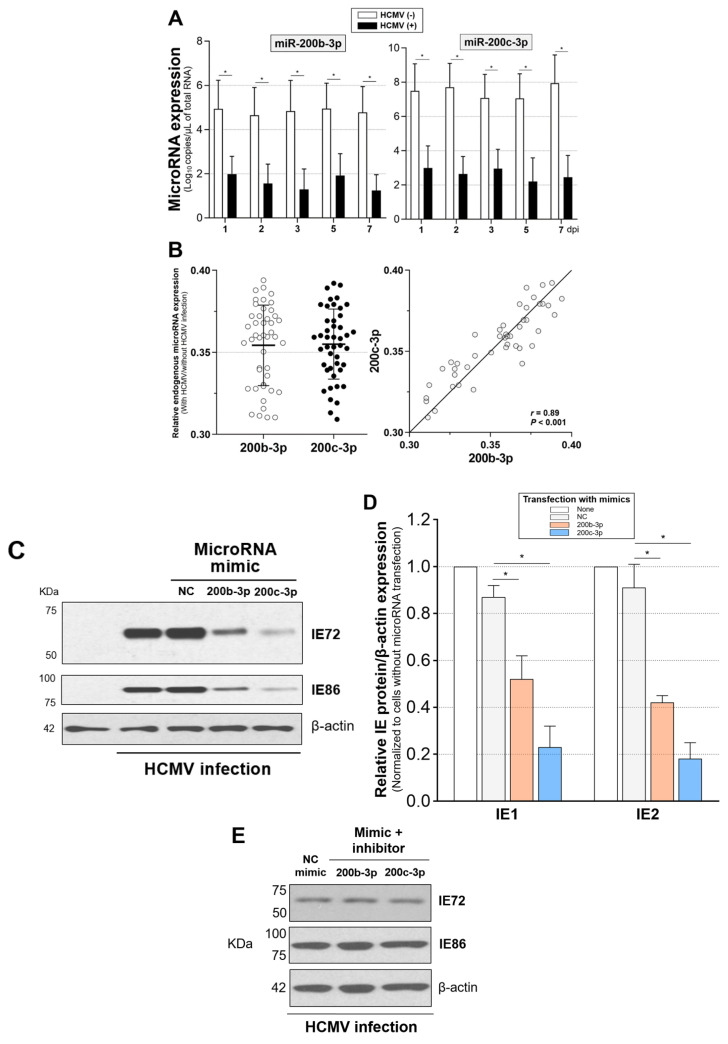
Downregulated expression of miR-200b-3p and miR-200c-3p in the background of acute HCMV infection was associated with IE72 and IE86 expression and HCMV replication. (**A**) Copies of endogenous miR-200b-3p and miR-200c-3p at 1, 2, 3, 5, and 7 days during HFF-1 cell culture with or without HCMV Towne strain (0.1 of MOI) measured using quantitative real-time reverse-transcription PCR (RT-PCR) with small RNAs isolated from pelleted cells (1 × 10^6^/well; 6-well plates). Absolute levels of miRNAs in independent standard curves for separate experiments are expressed as copies/µL of total RNA determined using real-time RT-PCR. Data are representative of triplicate measurements in three separate experiments for each miRNA per day. Expression of miR-200b-3p and miR-200c-3p in cells with and without HCMV infection (**B**, **left panel**) was estimated on days 1, 2, 3, 5, and 7 (total 45 measurements); then, correlations between them were determined (**B**, **right panel**). Middle and error bars indicate means and standard deviation, respectively, in the left panel. (**C**) HFF-1 cells (1 × 10^6^/well) were infected at 2 dpi with HCMV Towne strain at an MOI of 0.1 in 6-well plates and then transfected with miR-200b-3p, miR-200c-3p, or NC mimic (300 ng each). (**D**) The band intensity of IE72 and IE86 in immunoblots is expressed as fold value of cells that were not transfected with mimic (triplicate independent results from miRNA mimic-transfected HFF-1 cells). (**E**) Antisense inhibitor was transfected into HFF-1 cells 1 h after the simultaneous exposure to HCMV and mimic. Thereafter, the expression of IE72 and IE86 was investigated at 2 dpi. (**F**) Absolute HCMV titers in HFF-1 cells transfected with miR-200b-3p, miR-200c-3p, or NC mimic measured at 1, 2, 3, 5, and 7 dpi by measuring the expression of *UL83* using real-time PCR (left panel, cell lysates; right panel culture supernatants). HCMV viral load determined as log_10_ IU/µL in total DNA by PCR. Plot depicting HCMV titers represents triplicate measurements in three separate experiments per day. Upper lines in box or dot and error bars in (**A**,**B**,**D**,**F**) indicate means and SEM, respectively. * *p* < 0.001, compared with cells without HCMV infection (**A**) or transfected with NC mimic (**D**,**F**).

**Figure 4 ijms-23-02769-f004:**
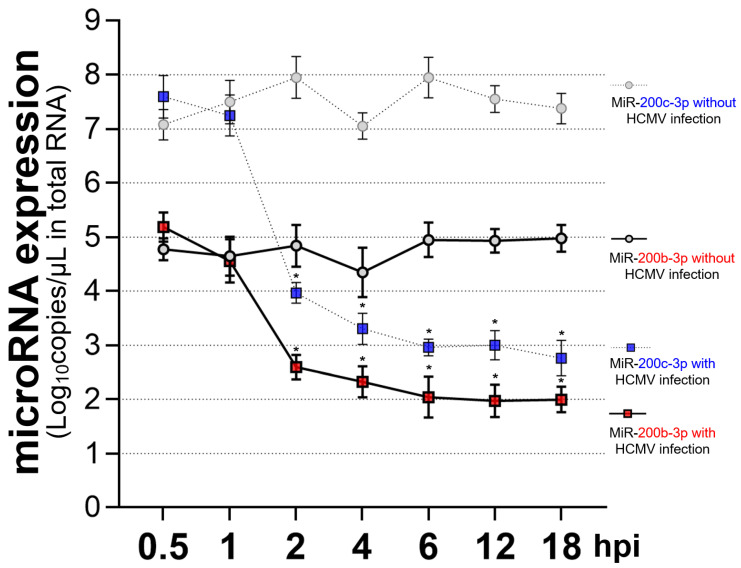
Dynamic changes in the expression of endogenous miR-200b-3p and miR-200c-3p during early lytic HCMV replication. Expression of miR-200b-3p and miR-200c-3p quantified using RT-PCR at indicated hpi in lysates of HFF-1 cells (1 × 10^6^/well) infected with or without the HCMV Towne strain at an MOI of 0.1. * *p* < 0.001, compared with cells without HCMV infection.

**Figure 5 ijms-23-02769-f005:**
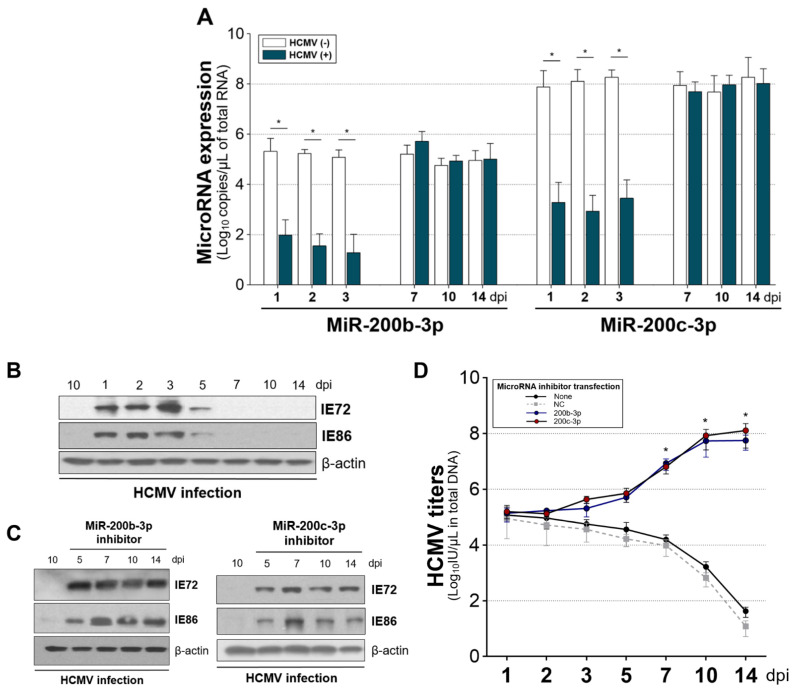
Deliberate downregulation of miR-200b-3p and miR-200c-3p during quiescent latency can induce IE72 and IE86 expression and HCMV replication.THP-1 cells infected with the HCMV Toledo strain at an MOI of 5 were cultured for 14 dpi and were harvested at the indicated dpi. (**A**) miR-200b-3p and miR-200c-3p expression measured by quantitative real-time PCR (triplicate measurements in three separate experiments daily). Levels of IE72 and IE86 protein were measured by immunoblotting HCMV-infected cells in the absence of miRNA mimic or inhibitor transfection (**B**) or with miR-200b-3p or miR-200c-3p inhibitors (**C**, left and right panels, respectively). (**D**) Titers of HCMV measured by real-time PCR using *UL83* primers. Quantitative results (**A**,**B**) of triplicate measurements in three separate experiments per dpi (*n* = 9) are shown as one box or one dot. Upper lines in the box or dots and error bars indicate means and SEM, respectively. * *p* < 0.001 vs. miRNA or HCMV DNA copies in cells without HCMV infection (**A**) and in NC mimic-transfected cells (**D**).

**Figure 6 ijms-23-02769-f006:**
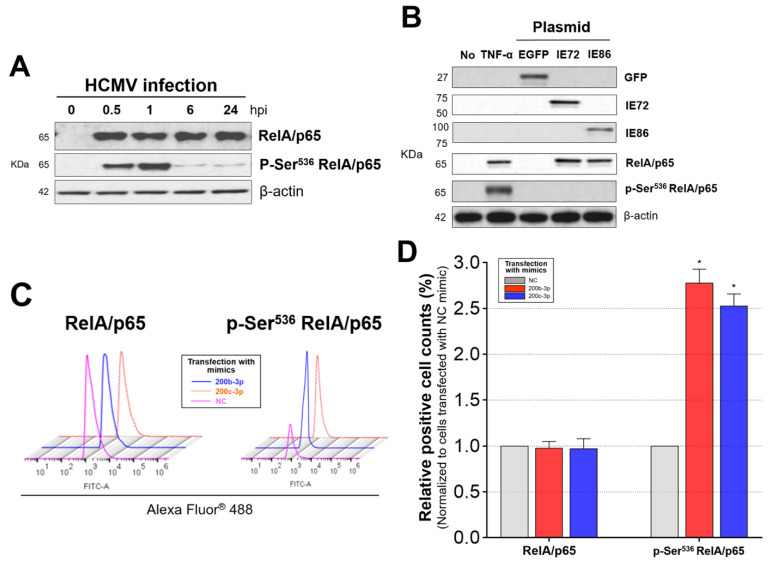
Phosphorylation of the RelA/p65 subunit on Ser^536^ is caused by IE proteins, and the downregulation of IE proteins in response to the upregulation of miR-200b-3p and miR-200c-3p is associated with increased levels of p-Ser^536^ RelA/p65. (**A**) The levels of RelA/p65 and p-Ser^536^ RelA/p65 proteins were measured at the indicated time points by immunoblotting. HFF-1 cells infected with the Towne strain of HCMV at an MOI of 0.1. (**B**) IE72 or IE86 plasmids were transfected into U373MG cells (1 × 10^6^/well; 6-well plates) that were lysed 2 days later. Total protein from pelleted cells was analyzed by immunoblotting using monoclonal antibodies (mAb) against IE72 (72 kDa), IE86 (86 kDa), RelA/p65 (RelA/p65, 65 kDa), phosphorylated RelA/p65 (Ser^536^) (p-Ser^536^ RelA/p65, 65 kDa), and β-actin. Negative and positive controls comprised cells transfected with EGFP plasmid and incubated with TNF-α (10 ng/mL) for 30 min, respectively. HFF-1 cells infected with the HCMV Towne strain and transfected with the miR-200b-3p, miR-200c-3p mimic or inhibitor, and NC mimic (each of 300 ng). HFF-1 cells that were p-Ser^536^ RelA/p65-positive were assessed by flow cytometry (**C**) and fluorescence images (**E**) obtained at 2 dpi. Cell counts were normalized to NC mimic-transfected cells (**D**). Fluorescence images without mAb were stained with polyclonal IgG secondary Ab. The levels of RelA/p65, p-Ser^536^ RelA/p65, and p105/p50 proteins were measured by immunoblotting in HFF-1 cells transfected with miR-200b-3p/miR-200c-3p mimics and infected with Towne strain (**F**) or with miR-200b-3p and miR-200c-3p inhibitors immediately thereafter (**G**). (**H**) Band intensity in immunoblots. Box plots show values obtained from triplicate experiments (**D**,**H**). Upper lines and error bars indicate means and SEM, respectively. * *p* < 0.001 vs. cells transfected with NC mimic.

**Figure 7 ijms-23-02769-f007:**
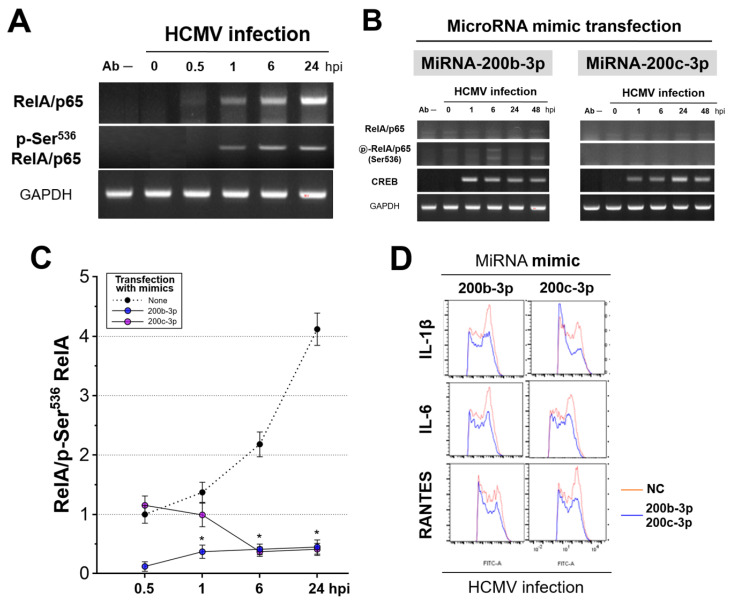
Nuclear factor−kappa B RelA/p65 phosphorylated at Ser^536^ bound to MIEP enhancer region and upregulated miR-200b-3p and miR-200c-3p led to increased p-Ser^536^ RelA/p65 binding to MIEP and decreased pro-inflammatory cytokine expression. We amplified the MIEP enhancer region by PCR in two cell groups after ChIP with RelA/p65 and p-Ser^536^ RelA/p65 mAbs. HFF-1 cells infected with HCMV Towne strain (MOI of 0.1) without (**A**) and with (**B**) miR-200b-3p or miR-200c-3p mimic transfection at indicated time points. (**C**) Band intensity of NF-κB RelA/p65 and p-Ser^536^ RelA/p65 is expressed as the ratio of amplified DNAs. (**D**) The expression of IL-1β, IL-6, and RANTES (CCL5) at 2 dpi in HFF-1 cells infected with the HCMV Towne strain and in those transfected with miR-200b-3p, miR-200c-3p, and NC mimic measured by flow cytometry. Dots and error bars indicate means and SEM, respectively. * *p* < 0.001 vs. cells without transfected microRNA mimic.

**Figure 8 ijms-23-02769-f008:**
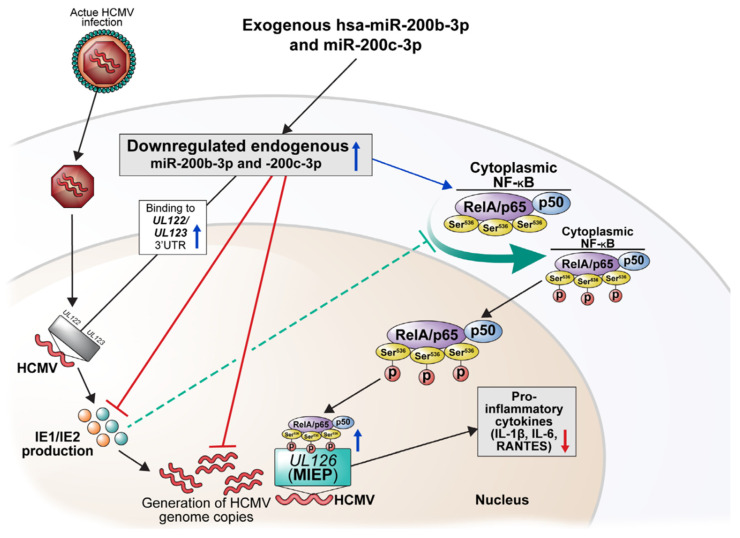
Schematic illustration of the potential benefits of exogenous miR-200b-3p and miR-200c-3p in acute lytic HCMV infection. Blue and red arrows or lines indicate the enhancing and inhibitory effects of exogenous miR-200b-3p and miR-200c-3p, respectively. Green dashed line indicates the direct suppression by IE proteins.

**Table 1 ijms-23-02769-t001:** PCR primers and oligonucleotides for plasmids.

PCR Primers		Sequence (5′→3′)	
HCMV Genes		Forward	Reverse
*UL122*		CAACATGATCATCCACGCTG	GAGACTTGTTCCTCAGGTCC
*UL123*		AAGCGGCCTCTGATAACCAAGCC	AGCACCATCCTCCTCTTCCTCTGG
*UL83*		TGCCCTGGATGCGATACTG	AGGACCTGACGATGACCG
MIEP	Upstream	GGTCAAAACAGCGTGGATGG	CGTGTACGGTGGGAGGTCTA
	Downstream	CCGTAAGTTATGTAACGCGGA	ACCGCCATGTTGACATTGATT
**Synthetic oligonucleotides ^a^**		**Sequence (5′→3′)**	
**Recombinant plasmids**			
Wild-type *UL123*-3′UTR-pEGFP		GAATTCACTATTGTATATATATATCAGTTACTGTTATGGATCCCACGTCACTATTGTATACTCTATATTATACTCTATGTTATACTCTGTAATCCTACTCAATAAACTCGAG	
Mutant *UL123*-3′UTR-pEGFP ^b^		GAATTCACTATTGTATATATATATCAGTTACTGTTATGGATCCCACGTCACTATTGTATACTCTACGCGCTACTCTATGTTATACTCTGTAATCCTACTCAATAAACTCGAG	
*UL122-123*- 3′UTR-pEGFP		CTCGAGTTTATTGAGTAGGATTACAGAGTATAACATAGAGTATAATATAGAGTATACAATAGTGACGTGGGATCCATAACAGTAACTGATATATATATACAATAGTGAATTCAGCCAGCAAGACAGCGATCTCGAGGTGAAAAACTGGAAAGAGAGACATGGACTCTTGTACATAGTGATTCCCCGTGACAGTATTAACGTGTGGTGAGAATGCTGTTTAATAAAAGAATTC	

^a^ Upper and lower underlines indicate XhoI and EcoRI restriction sites, respectively. ^b^ Bold font indicate altered oligonucleotides (wild-type sequence—TAATA).

## Data Availability

The data that support the findings of this study are available from the corresponding author.

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
