# Peer review of "Human MicroRNAs Attenuate the Expression of Immediate Early Proteins and HCMV Replication during Lytic and Latent Infection in Connection with Enhancement of Phosphorylated RelA/p65 (Serine 536) That Binds to MIEP"

_ijms, 2022, doi:10.3390/ijms23052769_

Round 1

Reviewer 1 Report

Hong et al. present a manuscript relating to human microRNAs that attenuate the expression of immediate early proteins (IE72 and IE86) of HCMV. They identify two hsa-miRs that are able to reduce the expression of immediate early proteins expressed from the MIEP. At the same time these hsa-miRs are downregulated at the beginning of the lytic infection allowing for high expression of immediate early proteins. They also show that IE72 and IE86 inhibit phosphorylation of RelA/p65 on S536, which can have effect on its function as a transcriptional regulator and inhibit pro-inflammatory cytokine production.

The manuscript presents interesting, relevant and novel data. It also contains quite substantial amount of data. I have only minor comments that I would like the authors to address:

  1. The connection for the set of experiments related to NFkB to the remaining part of the data is not highlighted sufficiently. A short sentence explaining the rationale for the presented experiments should be added at the beginning of section 2.6. Additionally, please add a clarification of the connection of the NFkB section (starting in line 596)  with the rest of the discussion section.
  2. Please extand the sentence in lines 581 and 582 (We and others...) clarifying the use of a luciferase reporter assay with MIEP promoter.
  3. line 588: please clarify - I believe the sentence should read something like that: ... inhibitors that counterbalanced the effects of exogenously supplied miRNA returned the expression of IE72 and IE86 to initial level.
  4. line 611: the data for direct inhibition of RelA/p65 phosphorylation on S536 by IE proteins has not been presented, therefore please modify this sentence accordingly.

Author Response

→ We do appreciate the reviewer’s detailed review and consideration for publication. According to the reviewer’s comments, we had tried to upgrade and revise the whole manuscript. Because we fully consent to the reviewer’s opinions, we described our responses/answers point-by-point for the following each comment.

Specific comments:

  1. 1. The connection for the set of experiments related to NF-kB to the remaining part of the data is not highlighted sufficiently. A short sentence explaining the rationale for the presented experiments should be added at the beginning of section 2.6. Additionally, please add a clarification of the connection of the NF-kB section (starting in line 596) with the rest of the discussion section.

→ According to this point, we inserted the below sentences in the results and discussion section, even though we thoroughly explained the connection for the set of experiments related to NF-kB to the expression of miR-200b-3p/-200c-3p in the introduction section (line: 80-96).

  • [Result – Section 2.6. Line: 419-422]: The NF-κB binding to MIEP can potentially and rapidly induce the activation of genes encoding the IE72 and IE86 during acute HCMV lytic infection. Therefore, there could be substantial relationship between the activation of NF-κB and production of IE proteins.

→ In addition, we moved the below sentence from introduction to the discussion section.

  • [Discussion – Line: 585-588]: The site-specific post-translational phosphorylation of RelA/p65, RelB, and c-Rel subunits after the degradation of inhibitor of NF-κB (IκBα) in the canonical NF-κB pathways plays an important role in modulating the downstream effects of NF-κB in response to a wide range of inflammatory stimuli [67,68,96,97].

  1. 2. Please extend the sentence in lines 581 and 582 (We and others...) clarifying the use of a luciferase reporter assay with MIEP promoter.

→ According to this comment, we added some words in the revised manuscript.

  • [Discussion – Line: 570-572]: We and others have also found that fluorescence emission is reduced by 50% in luciferase reporter assays with MIEP promoter [34,35].

  1. 3. line 588: please clarify - I believe the sentence should read something like that: ... inhibitors that counterbalanced the effects of exogenously supplied miRNA returned the expression of IE72 and IE86 to initial level.

→ The point of reviewer is exactly right. We added the phrase of “to initial level” to clarify this sentence.

  • [Discussion – Line: 576-578]: In addition, miR-200b-3p and miR-200c-3p inhibitors that counterbalanced the effects of exogenously supplied miRNA returned the expression of IE72 and IE86 to initial level.

  1. 4. line 611: the data for direct inhibition of RelA/p65 phosphorylation on S536 by IE proteins has not been presented, therefore please modify this sentence accordingly.

→ Our data showed that U373MG cells expressing IE72 and IE86 protein did not express p-Ser536RelA/p65, even though RelA/p65 was expressed in U373MG cells transfected with IE72 or IE86 plasmids (Figure 6B). These initial findings indicated that IE72 and IE86 inhibit the phosphorylation of RelA/p65 at Ser536, and that acute infection with HCMV does not induce RelA/p65 phosphorylation at Ser536 (unlike the case observed in the background of infection with other microbes.

Figure 6B. The plasmid expressing IE72 or IE86 directly inhibited the expression of p-Ser536 RelA/p65 (RelA/p65 phosphorylation on S536), but not unphosphorylated RelA/p65.

→ Please see the Figure 6B and the description of this data (2.6 section of results – Line: 427-439)

→ According to the reviewer’s comment and clarify the association between IE72/IE86 expression and RelA/p65 phosphorylation on S536 (p-Ser536 RelA/65), we deleted the words of “direct or directly in the whole revised manuscript.

  • [Abstract]: We determined that IE72/IE86 alone can directly inhibit the phosphorylation of RelA/p65 at the Ser536 residue and that p-Ser536 RelA/p65 binds to the major IE promoter/enhancer (MIEP).
  • [Results – heading of section 2.6, Line: 417-418]: 6. IE proteins and MiR-200b-3p or miR-200c-3p are directly associated with phosphorylation of RelA/p65 at Ser536
  • [Results – Section 2.6, Line: 436-438]: These initial findings indicated that IE72 and IE86 directly inhibit the phosphorylation of RelA/p65 at Ser536, and that acute infection with HCMV does not induce RelA/p65 phosphorylation at Ser536
  • [Results – Legend of Figure 6, Line: 491-492]: Figure 6. Phosphorylation of the RelA/p65 subunit on Ser536 is directly caused by IE proteins and downregulation of IE proteins in response to upregulation of miR-200b-3p and miR-200c-3p is associated with increased levels of p-Ser536 RelA/p65.
  • [Discussion – Line: 602-606]: This study on acute HCMV infection showed that the p-Ser536 RelA/p65 negatively affects IE protein levels and NF-κB activity by binding to MIEP. The expression of IE proteins is associated with directly inhibition of the phosphorylation of RelA/p65 at Ser536 and the levels of p-Ser536 RelA/p65 are decreased during acute lytic infection of HCMV when IE proteins are expressed.

Reviewer 2 Report

The authors wrote an interesting article.

The article is well written and of scientific interest 

I'd suggest in introduction to add  in 

Subclinical immune 37 senescence caused by intermittent reactivation of latent HCMV infection can induce 38 various chronic inflammatory morbidities and aging in individuals

a sentence or 2, with the difficultis in making a good diagnosis and that metagenomics can help in the diagnostic process. Please have a look and cite if you require so the following paper: PMID: 31276030 and PMID: 31312608 

Please improve the limitations of the study

Author Response

→ We do appreciate the reviewer’s detailed review and consideration for publication. According to the reviewer’s comments, we had tried to upgrade and revise the whole manuscript. Because we fully consent to the reviewer’s opinions, we described our responses/answers point-by-point for the following each comment.

Specific comments:

  1. I'd suggest in introduction to add in Subclinical immune senescence caused by intermittent reactivation of latent HCMV infection can induce various chronic inflammatory morbidities and aging in individuals a sentence or 2, with the difficulties in making a good diagnosis and that metagenomics can help in the diagnostic process. Please have a look and cite if you require so the following paper: PMID: 31276030 and PMID: 31312608.

→ According to this comment, we added the below sentence in the revised manuscript.

  • [Introduction – Line: 33-37] - Subclinical immune senescence caused by intermittent reactivation of latent HCMV infection can induce various chronic inflammatory morbidities and aging in individuals with the difficulties in making a good diagnosis and that metagenomics can help in the diagnostic process [1-7].

  • We searched and checked two papers (PMID: 31276030 and PMID: 31312608) in detail. Two review articles were dealing with the subject of metagenomics roles (especially, in microbiome studies) in ophthalmology field [PMID: 31276030 – “Metagenomics in ophthalmology: current findings and future prospectives, PMID: 31312608 – “Metagenomics in ophthalmology: Hypothesis or real prospective?]. There was no description of the cytomegalovirus. For this reason, we did not insert two paper in references. We ask for reviewer’s deep understanding of this decision.

  • Instead of the reviewer’s suggested articles, we cited the below papers, which presented that cytomegalovirus were isolated from bronchoalveolar lavage fluid (BALF) by metagenomic nextgeneration sequencing (mNGS) (mNGS-based diagnosis for CMV).

(Article: PMID: 31813078 – “Application of metagenomic nextgeneration sequencing for bronchoalveolar lavage diagnostics in critically ill patients. Eur J Clin Microbiol Infect Dis 2020;39: 369–74.” PMID: 31856779 – “Metagenomic next-generation sequencing for mixed pulmonary infection diagnosis. BMC Pulm Med. 2019 Dec 19;19(1):252”)

  • [New references]
  1. 6. Li, Y.; Sun, B.; Tang, X.; Liu, Y.L.; He, H.Y.; Li, X.Y.; Wang, R.; Guo, F.; Tong, Z.H. Application of metagenomic next-generation sequencing for bronchoalveolar lavage diagnostics in critically ill patients. J. Clin. Microbiol. Infect. Dis. 2020, 39, 369-374.
  2. Wang, J.; Han, Y.; Feng, J. Metagenomic next-generation sequencing for mixed pulmonary infection diagnosis. BMC Pulm. Med. 2019, 19, 252.

  1. Please improve the limitations of the study

→ According to the reviewer’s point, we newly added the sentences and references describing the limitation of our experiment in the revised manuscript based on the reviewer #2’s comment.

  • [Discussion – Line: 619-626]: Our experiments have the following limitation. There would be conceptual debates whether THP-1 cells represent a true HCMV latency model, even though many studies used THP-1 cells for an experimental latent-reactivation model [28,89,91,108,109]. It would be quite informative to evaluate the effect of miR-200b-3p/-200c-3p and their regulation in other models of latency as well as in normal human bone marrow cells (or more specifically CD34+ HPCs). If we could confirm the role of two microRNAs on HCMV lytic and latent infection from various human cells, we will be able to secure more evidence to apply the research results to humans.

  • [New references]
  1. 108. Arcangeletti, M.C.; Vasile Simone, R.; Rodighiero, I.; De Conto, F.; Medici, M.C.; Maccari, C.; Chezzi, C.; Calderaro, A. Human cytomegalovirus reactivation from latency: Validation of a "switch" model in vitro. J. 2016, 13, 179.
  2. 109. Gan, X.; Wang, H.; Yu, Y.; Yi, W.; Zhu, S.; Li, E.; Liang, Y. Epigenetically repressing human cytomegalovirus lytic infection and reactivation from latency in thp-1 model by targeting h3k9 and h3k27 histone demethylases. PLoS One 2017, 12, e0175390.

Reviewer 3 Report

Hong and co-authors present a concise and well-written article assessing the role of two human miRNAs (miR-200b-3p and miR-200c-3p) in controlling HCMV IE expression. This manuscript is well presented and contains the appropriate controls (NC mimic, dUL123 binding site in the UTR, ) and multiple methods (viral titers (cell and sup) as well as RNA and protein expression levels, etc) for the experiments. Other strengths of this paper include the use of a clinical strain of HCMV, analysis during actual infection, and the use of two different cell model systems. The data supports the conclusions drawn and is new and interesting data for how HCMV and cellular regulation interplay during infection. The data is presented well and the introduction and discussion appropriately center this data in the context of the field. I feel this is a strong manuscript and have only a few very minor comments for the authors to consider.

Very minor comments:

  • It is somewhat confusing (and less strong) that the authors state in the text: “45 measurements of lysates or supernatants within 7 days” (line 268), it was hard to understand what this meant in terms of replicates. However, the legend clearly states 3 independent experiments each with 3 replicates for 5 timepoints (also 45 measurements). It would make more sense to me to use this same phrasing in the text.
  • There are some font issues with the labeling in the figures (i.e. Figures 5, 6, 7 have some very large fonts in the version and panel letters are different sizes).
  • There are conceptual debates in the field as to whether THP-1 cells represent a true latency model, but the authors clearly state the systems they are using and provide the appropriate data for the models chosen. It would be interesting to see the effect of these miRNAs and their regulation in other models of latency, but this is outside the scope of this paper. It would be interesting to know if these miRNAs are commonly expressed in normal human bone marrow cells (or more specifically CD34+ HPCs) and therefore the authors could speculate that their data would translate in vivo, however, this would be a point for discussion only.

Author Response

Specific comments:

Very minor comments:

  1. It is somewhat confusing (and less strong) that the authors state in the text: “45 measurements of lysates or supernatants within 7 days” (line 268), it was hard to understand what this meant in terms of replicates. However, the legend clearly states 3 independent experiments each with 3 replicates for 5 time points (also 45 measurements). It would make more sense to me to use this same phrasing in the text.

→ We fully consent to this point. Along with the reviewer’s comment, we changed the obvious phrases for both figure 3B and 3F as the below in the text.

  • [2. Results – 2.4. section, Line: 267 and 269 for description about Figure 3B, left and right panel]: data are from nine measurements each on days 1, 2, 3, 5, and 7
  • [2. Results – 2.4. section, Line: 292-294 for description about Figure 3F]: three independent measurements each with three replicates of lysates or supernatants each at 1, 2, 3, 5, and 7 dpi

  1. There are some font issues with the labeling in the figures (i.e. Figures 5, 6, 7 have some very large fonts in the version and panel letters are different sizes).

→ According to the reviewer’s comment, we adjusted and revised the sizes and labeling of font and panel letters to be consistent in the revised Figures (3A, 3B, 3F, 5D, 6E, 6F, 6G, 7A/B/C/D) . Please see the revised manuscript.

  1. There are conceptual debates in the field as to whether THP-1 cells represent a true latency model, but the authors clearly state the systems they are using and provide the appropriate data for the models chosen. It would be interesting to see the effect of these miRNAs and their regulation in other models of latency, but this is outside the scope of this paper. It would be interesting to know if these miRNAs are commonly expressed in normal human bone marrow cells (or more specifically CD34+ HPCs) and therefore the authors could speculate that their data would translate in vivo, however, this would be a point for discussion only.

→ We fully consent to this important point. We eagerly searched the papers or studies for evaluating the expression of miR-200b-3p/-200c-3p in normal human bone marrow cells (especially, CD34+ hematopoietic stem cells), but we did not find the studies corresponding to this issue. In addition, the reviewer’s valuable suggestion requires sufficient further researches from human sample. Consistent with the reviewer’s opinion, we think that the human studies, particularly from bone marrow of health volunteers, are outside the scope/aim/main purpose of our experiments performed with in vitro cell culture. We ask for the reviewer’s full understanding of our thoughts.

→ We described this valuable issue as the limitation of our research in the revised discussion section. The reviewer #2 also commented the improvement of study limitation.

  • [Discussion – Line: 619-626]: Our experiments have the following limitation. There would be conceptual debates whether THP-1 cells represent a true HCMV latency model, even though many studies used THP-1 cells for an experimental latent-reactivation model [28,91,93,108,109]. It would be quite informative to evaluate the effect of miR-200b-3p/-200c-3p and their regulation in other models of latency as well as in normal human bone marrow cells (or more specifically CD34+ HPCs). If we could confirm the role of two microRNAs on HCMV lytic and latent infection from various human cells, we will be able to secure more evidence to apply the research results to humans.

  • [New references]
  1. 108. Arcangeletti, M.C.; Vasile Simone, R.; Rodighiero, I.; De Conto, F.; Medici, M.C.; Maccari, C.; Chezzi, C.; Calderaro, A. Human cytomegalovirus reactivation from latency: Validation of a "switch" model in vitro. J. 2016, 13, 179.
  2. 109. Gan, X.; Wang, H.; Yu, Y.; Yi, W.; Zhu, S.; Li, E.; Liang, Y. Epigenetically repressing human cytomegalovirus lytic infection and reactivation from latency in thp-1 model by targeting h3k9 and h3k27 histone demethylases. PLoS One 2017, 12, e0175390.
